# The Impact of Obesity on the Fitness Performance of School-Aged Children Living in Rural Areas—The West Attica Project

**DOI:** 10.3390/ijerph191811476

**Published:** 2022-09-12

**Authors:** Charilaos Tsolakis, Evgenia D. Cherouveim, Apostolos Zacharias Skouras, Dimitrios Antonakis-Karamintzas, Cara Czvekus, Panagiotis Halvatsiotis, Olga Savvidou, Panagiotis Koulouvaris

**Affiliations:** 11st Department of Orthopedics Surgery, Medical School, National and Kapodistrian University of Athens, 12462 Athens, Greece; 2Sports Performance Laboratory, School of Physical Education & Sports Science, National and Kapodistrian University of Athens, 17237 Athens, Greece; 3School of Physical Education & Sports Science, National and Kapodistrian University of Athens, 17237 Athens, Greece; 4Health Science, James Madison University, Harrisonburg, VA 22807, USA; 52nd Department of Internal Medicine Propaedeutic “Attikon” University Hospital, Medical School, National and Kapodistrian University of Athens, 12462 Athens, Greece

**Keywords:** pediatric obesity, childhood obesity, overweight, prevalence, elementary school, physical fitness, remote health

## Abstract

This study aimed to investigate the relationship of body mass index (BMI) with muscle and cardiorespiratory fitness in children living within rural areas (regional unit of West Attica) in Greece. Participants included 399 students (187 boys, 212 girls), ages 8–12 years old, and were evaluated in physical performance tests. The point prevalence of overweight and obesity was 21.39% and 26.20% in boys, and 19.90% and 23.79% in girls. Significant differences were observed in all physical performance tests (handgrip, long jump, shuttle run, trunk flexors, and extensors endurance) between normal weight and obese participants. BMI was positively correlated with handgrip (r = 0.442−0.462, *p* < 0.001). There was a negative association with long jump (r = −0.206, *p* < 0.001), 20 m shuttle run (r = −0.394, *p* < 0.001), trunk flexors (r = −0.403, *p* < 0.001) and trunk extensors endurance (r = −0.280, *p* < 0.001). The regression analysis showed that 20–30% of the overall variation for physical performance assessments could be accounted for by BMI, age, and sex. With the exception of the long jump and the endurance of the trunk extensors, BMI alone may explain more than 10% of the outcome of most tests. This study highlights the determinant of BMI on muscle and cardiorespiratory fitness. The management of obesity should begin early in childhood to prevent adult chronic cardiovascular and metabolic diseases.

## 1. Introduction

The rate of childhood obesity has substantially increased worldwide during the last several decades [1]. This increase has led many researchers to characterize this rise as a global public health issue and to call for action [1,2,3]. Childhood obesity has also dramatically increased over the last few decades in Greece. In 2019, the point prevalence of overweight and obesity in Greek elementary school children was more than 40% [4]. Although overweight and obesity prevalence in children and adolescents has reached a plateau in high-income countries [5], Greece is still considered one of the highest worldwide [6,7]. Among European countries where one of three children is overweight or obese, Greece has the highest prevalence [4,8,9]. Recently, the World Obesity Federation estimated an annual increase of 2.1% in child obesity between 2010–2030, putting Greece at a high rate of increase [10].

The health impact of obesity and low physical activity levels in childhood is well documented through epidemiological studies [11,12]. Childhood obesity is associated with cardiovascular and metabolic diseases [13] and is also strongly associated with increased obesity rates in adulthood [14,15]. The likelihood of becoming obese as an adult is approximately five times higher in obese children and adolescents than in non-obese [16]. However, a recent meta-analysis indicates that 10–15% of non-obese children will be overweight or obese as middle-aged adults [16]. A 4-year cohort study of 19,504 Chinese children, aged 6–9, identified a reciprocal relationship between body mass index (BMI) and physical performance. In particular, prospectively, normal weight was related to higher physical fitness, and vice versa [17].

Physical fitness and activity level positively correlate with various health-related outcomes in school-aged children [18,19,20,21]. A higher level of physical activity during childhood (4–11 years old) is correlated with less body fat [18], reduced anxiety and depressive symptoms [19], and improved cardiorespiratory fitness and muscular strength [18] by the time of early adolescence [20]. Similarly, physical fitness during childhood is positively correlated with academic performance [22], skeletal health [23] and body composition [24] and is inversely associated with anxiety [25] and cardiovascular diseases at a later stage [23]. Physical fitness in school-aged children also positively predicts self-rated health in early adulthood, independent of sex [26]. A recent systematic review highlighted that well-designed school-based intervention programs for promoting physical activity and fitness might have significant promise for preventing obesity [27].

Obesity is associated with reduced physical activity levels and fitness. Interestingly reduced physical activity and increased sedentary time are predicted by obesity, but not the other way around [28]. When studies used body mass to assess obese children, they revealed a lower level of maximal oxygen consumption or cardiovascular fitness compared to normal-weight children. While studies assessing fat-free mass reported comparable fitness levels between obese and normal-weight children [29]. Furthermore, severe obesity (BMI > 40 kg/m^2^) is associated with an important decrease in physical activity levels, exercise performance [30], and motor skills [31]. Compared to boys, although girls the same age tend to be more obese and less active, with a lower participation rate in organized physical activities programs [32,33], studies in the Greek child population have revealed a reverse relationship. Particularly in Greece, boys have a higher prevalence of being overweight and obese and have lower physical fitness performance and activity levels [31,34,35]. However, boys still present a higher participation rate in organized physical activity programs [34].

Limited studies in Greek populations have assessed the prevalence of childhood obesity with the association of physical fitness performance in rural areas. In a recent study on remote and isolated Greek islands, obese children aged 5–12 represented 23.8% and 13.2% of boys and girls. Overweight and obese children demonstrated lower performance in weight-bearing activities, such as the 20 m shuttle run test and standing long jump, compared to normal and underweight children; however, upper body muscle strength was higher [31]. Similarly, compared to children on the mainland, those residing on the islands had significantly higher rates of obesity (10.1 vs. 8.0%) and overweight (23.0 vs. 21.8%) [35]. Except for the speed test, children from the islands performed considerably worse than those from the mainland (25th percentile of age- and sex-specific normative values) on four fitness tests (20 m shuttle run test, maximum 10 × 5 shuttle run test, sit-ups test in 30 s, standing long jump) [35].

Among many factors (age, sex, physical inactivity, sedentary lifestyle, and genetics), area of residence and low socioeconomic individuals have a higher risk of experiencing childhood obesity [36]. In Greece, childhood obesity spanning from ages 5–12 years old is well-documented in remote islands, isolated islands, and rural and urban areas [6,31,37]. In the US and European countries, low socioeconomic status (SES) relating to high levels of childhood obesity was studied [38,39]. This association was also confirmed in Greece, where childhood obesity predictors were the mother’s age, parental BMI classification, and father’s type of occupation [40]. Children with low SES tend to be significantly less active than those in middle and high socioeconomic backgrounds [34,36,41]. Particularly after the economic crisis, Greece has a lot of suburban and rural areas with low SES, such as West Attica. West Attica is one of the 74 regional units of Greece and is located near Greece’s capital, Athens. In 2021, the population of western Attica was 164,864, with a population density of 160 inhabitants per km^2^. West Attica is subdivided into five municipalities: Aspropyrgos, Elefsina, Fyli, Mandra-Eidyllia, and Megara. West Attica is an inner-city with a high rate of immigrants and refugees, mainly from the former Soviet Union.

There is a scarcity of studies associating BMI and fitness levels in primary school children (8–12 years old) living in rural areas of mainland Greece. The purpose of this study was to evaluate the associations between BMI and physical fitness specific to muscle power of the lower and upper body strength, trunk muscle endurance, and aerobic endurance for children living in rural areas (West Attica) in Greece. We hypothesized that the higher the BMI, the lower the physical performance would be in children living in rural areas of Greece, specifically in the regional unit of West Attica.

## 2. Materials and Methods

### 2.1. Participants

The participants of the study were school-aged students living in municipalities of West Attica, a region characterized by a high unemployment rate, low-income families, and low educational status. After getting permission from the Hellenic Ministry of Education (ethical approval number Φ3/189116/Δ1, 2 December 2019), we contacted the school principals of the regional unit of West Attica. We had initial permission to access 11 elementary schools in January 2020, but due to COVID-19 pandemic school closures, we finished the study prematurely in March 2020. In total, 399 students (187 boys and 212 girls) from 4 schools aged 8–12 volunteered to participate in this study. All students have received medical clearance to participate in the study. A written parental consent form was collected from all participants following detailed information about the experimental procedure, potential benefits, and risks. Ministry of education approved the execution of the study (approval number Φ3/189116/Δ1) for the Review Board of 1st Orthopedics Department, School of Health Sciences, National and Kapodistrian University of Athens, and all the procedures were in accordance with the Helsinki Declaration of 2000 revision.

### 2.2. Procedure

Students were assessed in anthropometrical and physical performance tests in separate physical education (PE) sessions. All assessment tests took place at school yards from four PE teachers and one physiotherapist (see Acknowledgments). Apart from the 20 m shuttle run, all tests were performed individually. Before starting the physical performance tests, students performed a 10 min standardized warm-up (5 min jogging at their own pace and 5 min dynamic stretching). The physical performance tests were performed in a random sequence. Each test was explained and demonstrated before the data collection. Qualified PE teachers conducted all the evaluations.

#### 2.2.1. Anthropometrics

The anthropometric measurements were conducted in the first PE session. Body mass, standing height, and sitting height were measured to the nearest 0.1 kg (SECA 770, Hamburg, Germany) and 0.5 cm (SECA 213 Stadiometer, Hamburg, Germany). BMI was calculated as follows, BMI = body mass in kilograms divided by height in meters squared. Students were classified as underweight, normal-weight, overweight, and obese based on the age and sex BMI cut-off points of World Health Organization (WHO) 2007 norms [42].

#### 2.2.2. Physical Performance Testing

In the second PE session, students were evaluated in a series of physical performance trials consisting of long jump, trunk muscles endurance test (trunk extensors and flexors), handgrip, and multi-stage 20 m shuttle run test. Two measurements were recorded for the long jump and the handgrip, with the best value used for further analysis. Trunk muscle endurance and multi-stage 20 m shuttle run test were performed once. Verbal encouragement occurred during the entire test to help students achieve optimal performance.

#### 2.2.3. Assessment of Muscle Trunk Endurance

Abdominal muscle endurance was assessed with the trunk flexor endurance test [43]. Students sit on a mat with their knees bent approximately at 90 degrees. Another student secured their feet on the floor and arms crossed over their chest. The trunk was approximately 60 degrees away from the ground. Students were requested to stay in that position for as long as possible. The examiner informed the student when to start and simultaneously started the timer. On the student’s hip, a goniometer was placed with one angle in contact with the mat and the other angle on the student’s back to determine the angle of 60 degrees. Trunk extensor muscle endurance was assessed via the back-endurance test (Biering–Sørensen test) [44,45]. The back-endurance test is simple to perform and requires no additional equipment [45]. The student laid down in a prone position on the edge of a bench or desk and another student secured their feet. Again, the examiner informed the student when to start and simultaneously started the timer. The students were instructed to maintain a flat back. Both tests were finished when the student was (a) not able to correctly hold this position, (b) not able to hold this position anymore, or (c) reached 240 s. The participants were encouraged to maintain steady breathing during the test [46]. The trunk flexor and back-endurance tests have already been examined in elementary school children [47,48].

#### 2.2.4. Handgrip

A handheld dynamometer was used to measure the maximal isometric handgrip strength. Handgrip strength was measured using a portable hydraulic dynamometer (Jamar 5030J1, Horsham, PA, USA). The test involved participants standing with their testing forearm at a 90 degrees angle to the upper body, keeping it in a vertical position. Before the performance, we ensured the wrist and the forearm were in a mid-prone position and we explained maximal effort must be given. The testing protocol consisted of 3 maximal isometric contractions (3 s) for each hand with a 1 min rest between trials. The top scores from each hand were recorded in kilograms (kg). The best for right and left were considered for further analysis [49]. The intraclass correlation coefficient (ICC) for test–retest reliability for the handgrip test was 0.93 (*p* < 0.001).

#### 2.2.5. Standing Long Jump

Lower body muscle power was assessed with a standing long jump test. Subjects stood on a standardized starting point with their legs parallel and feet shoulder-width apart. They were allowed to perform two trials with 60 s rest before their measured attempts. Participants were instructed to bend their knees (the depth of the flexion was self-selected) and bring their arms behind the body. Then, with a powerful drive, they extended their legs, moved their arms forward, jumped as far as possible, and landed on both feet. The closest landing point to the starting line was recorded to the nearest centimeter. A minimum of 30 s rest was allocated between each jump. Two measurements were recorded, and the best trial was used for further analysis. All trials were measured to the nearest 0.01 m [50]. The ICC test–retest reliability for the long jump assessment was 0.92 (*p* < 0.001).

#### 2.2.6. 20 m Shuttle Run

Cardiovascular fitness was assessed through 20 m shuttle running. Participants were instructed to run between two lines 20 m apart following a sound signal emitted from an audiocassette. The frequency of signals increased by 0.5 km/h each minute from a starting speed of 8.5 km/h. The test was terminated when subjects could not maintain the prescribed pace for three consecutive signals. The equivalent shuttle running speed was used as an endurance performance indicator [51].

### 2.3. Data Analysis

Results are presented as means ± standard deviations. Data were analyzed with IBM SPSS Statistics v.25 software (IBM Corporation, Armonk, NY, USA). The normality of data was examined using the Kolmogorov–Smirnov test. A non-parametric Kruskal–Wallis test for independent samples was used to evaluate differences between sex across age groups (8, 9, 10, 11, and 12 years old) for anthropometric characteristics and exercise performance in physical fitness tests. A non-parametric Kruskal–Wallis test was used to identify differences between sex and BMI categories (underweight, normal, overweight, and obese) for all the selected physical fitness measures. Group comparison was made to examine the differences within the group. The differences between the underweight and normal group (S1) and the overweight and obese group (S2) were tested using the independent-samples Mann–Whitney U test. Spearman’s correlations coefficient was calculated to assess the relationship between anthropometrics (body mass, standing height, sitting height, BMI) and long jump, trunk muscles endurance (trunk extensors and flexors), handgrip, and multi-stage 20 m shuttle run test. The absolute value for Spearman’s rho was determined as weak (0.3), moderate (0.5), strong (0.8), and absolute correlation (1.0). The calculation of the intraclass correlation coefficient (ICC) with a 2-way mixed model was used to test–retest the reliability for the dependent variables handgrip and standing long jump.

Effect sizes were calculated for each comparison among groups. For the Mann–Whitney test, the formula was r^β^ = Ζ/√n, where Z = standardized test statistic, n = total sample size. For the Kruskal–Wallis test, the formula was η^2^ = H − k + 1/N − k, where H = the value obtained in the Kruskal–Wallis test (the Kruskal–Wallis H-test statistic), k = the number of groups, n = the total number of observations [52]. Effect size was determined as small for r = 0.1 and η^2^ = 0.01, medium for r = 0.3 and η^2^ = 0.059, and large for r = 0.5 and η^2^ = 0.138. Hierarchical linear regression analysis was performed. To examine relative contributions to each dependent variable separately (standing long jump, trunk muscles endurance, handgrip strength, and multi-stage 20 m shuttle run), independent variables were entered in blocks as follows: BMI in the first step, age in the second step, and sex (as dummy variable) in the third step. Autocorrelation in the residuals was tested with the Durbin–Watson statistic, setting an acceptable range between 1.50–2.50. Multicollinearity was examined using the tolerance and variance inflation factor (VIF). For each analysis, the statistical significance was set at *p* < 0.05.

The power of the sample size was calculated using G*Power 3.1.9.7 for Windows (Heinrich-Heine-Universität Düsseldorf, Düsseldorf, Germany). The analysis was conducted for each dependent variable which tested physical fitness. To compute achieved power between groups S1 (underweight and normal-weight) and S2 (overweight and obese), a post-hoc power analysis was used. Given alpha equals 0.05 and the actual effects sizes and samples sizes for each variable, the power of the sample size between groups was: 88.89% for dominant handgrip, 81.75% for non-dominant handgrip, 34.96% for long jump, 69.68% for trunk extensors endurance, and 18.33% for 20 m multi-stage shuttle run test. Power was near the maximum value of 1 for the trunk flexors’ endurance.

## 3. Results

### 3.1. Anthropometric Characteristics

Stature height (*p* < 0.001), sitting height (*p* < 0.001), and body mass (*p* < 0.001) were significantly different across age groups (Table 1). Group comparison across age groups for boys and girls revealed statistically significant differences; boys 8 to 11 years old and girls 9 to 11 years old differed in standing height; boys 9 to 11 years old boy in sitting height; and boys 9 to 10 years old and girls 8 to 11 years old in body mass. Anthropometric characteristics were similar (*p* > 0.05) between boys and girls across age groups except for sitting height, in which 9-year-old girls were shorter than boys. BMI was similar across age groups (8–12 years old) in both sexes. The prevalence for underweight boys and girls was 0.54% and 2.9%. Normal weight was 51.87% and 53.4%, overweight was 21.39% and 19.4% and obese was 26.20% and 24.2% for boys and girls.

### 3.2. Physical Performance Differences between BMI Groups

Exercise performance was significantly different across BMI groups (Table 2). Obese children of both sexes had the lowest performance in all physical fitness tests except for handgrip strength compared to normal-weight children. Compared to the overweight group, the obese group had a higher handgrip value and lower trunk extensor endurance. We found a sign of sex’s main effect on physical performance tests across BMI groups. Normal-weight girls had lower handgrip strength and aerobic capacity but greater trunk extensor strength than boys. Overweight girls had higher trunk endurance and lower aerobic capacity than overweight boys. The only difference between sexes in the obese group was a greater trunk extensor endurance in girls than boys. Large effect sizes were demonstrated between BMI groups for both sexes in all variables, except for handgrip strength of both hands and sexes and standing long jump for girls, which presented a moderate effect size.

### 3.3. Physical Performance Test Association with the Underweight and Normal Groups (S1) versus the Overweight and Obese Groups (S2)

We did not observe any differences in physical fitness tests between the underweight and normal weight groups. We plotted the underweight and normal-weight children in the first subgroup (S1) and compared the exercise performance with the second subgroup (S2), which consisted of overweight and obese children (Table 2). Girls in the S2 subgroup had greater handgrip performance but lower trunk flexors endurance compared to the S1 subgroup. The boys in S2 had lower jump ability and trunk flexor strength than in S1. Physical performance tests were similar between boys and girls. Effect size was moderate-to-large only for trunk flexors endurance in boys (η^2^ = 0.443, *p* < 0.001) and moderate in girls (η^2^ = 0.300, *p* < 0.001). Effect sizes for all the other variables were small for both sexes.

### 3.4. Correlations and Percentiles

We found statistically significant correlations between BMI and some of the physical performance tests, such as handgrip strength in both hands, long jump, shuttle run, trunk flexors, and extensors values (Table 3). A moderate-to-negative correlation was observed for trunk flexor endurance for both sexes. A weak-to-moderate positive correlation between handgrip strength for both hands and sexes was also revealed. Percentiles of exercise performance in physical fitness tests are shown in Table 4 for boys and Table 5 for girls.

### 3.5. Regression Analysis: BMI, Age, and Sex as Predictors for Physical Fitness

Regression analysis revealed that all models were statistically significant for each dependent variable. The first model (BMI as predictor) was found to significantly fit with all dependent variables (*20 m multi-stage shuttle run*: *p* < 0.001, R = 0.367, R^2^ = 0.135, adj. R^2^ = 0.131; *dominant handgrip*: *p* < 0.001, R = 0.417, R^2^ = 0.174, adj. R^2^ = 0.171; *non-dominant handgrip*: *p* < 0.001, R = 0.386, R^2^ = 0.149, adj. R^2^ = 0.147; *long jump*: *p* < 0.001, R = 0.232, R^2^ = 0.054, adj. R^2^ = 0.051; *trunk flexors endurance*: *p* < 0.001, R = 0.425, R^2^ = 0.181, adj. R^2^ = 0.178; *trunk extensors endurance*: *p* < 0.001, R = 0.277, R^2^ = 0.077, adj. R^2^ = 0.074).

Model 2 (BMI and age as predictors) was also found to be statistically significant in all dependent variables (*20 m multi-stage shuttle run*: *p* < 0.001, R = 0.536, R^2^ = 0.287, R^2^ change = 0.152, adj. R^2^ = 0.281; *dominant handgrip*: *p* < 0.001, R = 0.650, R^2^ = 0.423, R^2^ change = 0.250, adj. R^2^ = 0.420; *non-dominant handgrip*: *p* < 0.001, R = 0.639, R^2^ = 0.408, R^2^ change = 0.260, adj. R^2^ = 0.405; *long jump*: *p* < 0.001, R = 0.514, R^2^ = 0.265, R^2^ change = 0.211, adj. R^2^ = 0.261; *trunk flexors endurance*: *p* < 0.001, R = 0.461, R^2^ = 0.213, R^2^ change = 0.032, adj. R^2^ = 0.209; *trunk extensors endurance*: *p* < 0.001, R = 0.380, R^2^ = 0.145, R^2^ change = 0.068, adj. R^2^ = 0.140).

Model 3, where sex was entered as an added predictor variable, demonstrated a statistically significant fit (ANOVA *p* < 0.001, for all dependent variables). Statistical significance of R^2^ change was reached for all, except for handgrip strength (dominant and non-dominant hand). Sex did not contribute significantly on the prediction of the handgrip strength, adding on BMI and age (dominant: *p* = 0.177, R^2^ change = 0.003; non-dominant: *p* = 0.498, R^2^ change = 0.001). The proportion of the variance was explained in Model 3 by 31.3% for the 20 m multi-stage shuttle run test, 27.4% for the long jump, and 22.4% and 20.7% for the trunk flexor and extensor endurance.

Regression equations for each dependent variable are the following:20 m multi-stage shuttle run test (shuttles) = 10.525−1.786 × BMI + 5.284 × age + 5.053 × sex
Handgrip (dominant) (kg) = −10.382 + 0.352 × BMI + 2.124 × age
Handgrip (non-dominant) (kg) = −9.854 + 0.303 × BMI + 2.086 × age
Standing long jump (cm) = 73.709−2.007 × BMI + 9.504 × age + 4.532 × sex
Trunk flexors endurance (seconds) = 220.113 − 8.415 × BMI + 11.640 × age − 15.099 × sex
Trunk extensors endurance (seconds) = 107.484 − 5.435 × BMI + 15.866 × age − 32.769 × sex
where variable sex is 0 for girls and 1 for boys and age is demonstrated in years.

## 4. Discussion

This study investigated the relationship between BMI and physical performance in young school-aged children (8–12 years old) living in the regional unit of West Attica. The study’s main finding was that for school boys and girls ages 8–12 years old, BMI was positively correlated with handgrip strength and negatively with all weight-bearing activities. The results of our study also indicate that the point prevalence of obesity was 26.20% and 23.79% for boys and girls in West Attica, Greece. Overall, the overweight and obesity rate is more than 40% for both sexes. We can confirm the correlation between high BMI and low physical performance based on the findings. The regression analysis revealed that BMI, age, and sex could explain 20–30% of the total variance for physical performance tests. As the only predictor, BMI explains more than 10% of most test values, except for long jump and trunk extensor endurance.

Childhood obesity is a major public health challenge of the 21st century and has become a public health problem in many countries worldwide [2,50,53,54]. It is suggested that early childhood years provide the best opportunity to modify environmental risk factors of obesity [55]. Regarding the Greek population, although previous studies have examined childhood obesity and indicated 50% increments [6,56,57], more recent data revealed a significant decrease of 10% over the last decade [58]. Overweight and obesity in rural Greek areas have recently been evaluated and reported to have a prevalence of more than 30% [31,35,37]. In this study, we found the prevalence of being overweight was 21.39% and 19.90% for boys and girls, while the obesity rates were 26.20% and 23.79%. These rates are of the highest reported for the Greek student population, confirming reports from a prior demographic study, the total Greek student population represents approximately 40% [56]. The prevalence of being overweight and obese in the student population living in West Attica (a rural area) is higher than those reported for living on islands (30.1%) and the mainland (29.8%) [35]. Compared to our study, the same tendency, with boys being overweight and obese at a higher rate than girls, was observed in a recent survey among 3816 Greek school-children aged 11–13. The prevalence of being overweight and obese was 31.4% for boys and 18% for girls, being much lower than our findings, especially for girls [59]. This difference may be due to the different age groups tested in the two studies.

Childhood obesity seems to be multifaceted [13]. Several underlying mechanisms were proposed, including low SES, physical inactivity, and low physical fitness. SES is related to the risk of childhood obesity [41,55,60,61]. Recently, Makri et al. [59] highlighted a clear negative association between family affluence and being overweight and obese. A large portion of childhood obesity, 71%, is due to family factors including low parental income, poor diet, and limited access to physical activity. These factors can increase the vulnerability of becoming overweight or obese [60]. Although obesity is related to SES, differentiated from sex, age, and country low SES families; living in industrialized countries increases the possibility of adopting unhealthy eating habits and being obese [61,62]. High SES households tend to have children and adolescents with better physical fitness and a lower likelihood of having low or poor fitness [63]. The West Attica region is an industrial zone with a high unemployment rate, low-income households, and low parental educational status [64]. The residents in rural and remote areas have limited access to parks, playgrounds, and sports facilities [34]. The high prevalence of obesity suggests the need for increased active lifestyle habits among these communities by promoting fitness and healthy lifestyles to school-aged children.

The second determinant of childhood obesity is physical inactivity. Although our study did not measure daily physical activity level, there is a correlation between physical fitness and physical activity level [65]. It is well-established that there is a negative correlation between BMI and several physical fitness parameters in children living in rural areas [66]. In the last few decades, a few studies have reported that most children are inactive and have increased their sedentary behaviors [67,68,69].

The present study reported that overweight and obese children indicated lower exercise performance than normal and underweight children in all fitness tests, except for the handgrip strength in children between 9 and 11 years old. In 2018, Koulouvaris et al. [31] reported similar overweight and obesity rates, and low physical fitness levels in children living on remote and isolated Greek islands. Similarly, Arnaoutis et al. [35] found low physical fitness levels and an increased obesity rate in a large cohort of school-aged children from rural and urban Greek areas. Our study’s results were found to have higher overweight and obesity prevalence, similar to Tsimeas et al.’s findings [70]. This suggests that the location of residency has no obvious relationship to the prevalence of childhood obesity. This discrepancy may be attributed to methodological differences, such as the number and age of the participants, and mainly the determination of rural areas (remote areas, industrial zones, islands–mainland). It is important to mention that low physical fitness may be the result rather than the cause of obesity in school-aged children. Obese children may refrain from intense physical exercise due to the fear of reduced performance, thus further declining their fitness levels [31,71,72].

It is widely reported that school-aged children have progressively worse physical fitness [73]. Due to limited time for a physical education class, leisure activities, adapting unhealthy dietary habits, and increased sedentary behavior (exposure to screens) resulting in excess body weight [13,74]. Obesity is also associated with decreased physical and motor abilities in school-aged children [72,75,76], specifically those living in rural areas [31,35,56].

Although BMI does not measure overall fat or lean body mass, it is a simple and reliable indicator of obesity [77]. Cut-offs based on WHO documentation are considered a strong assessment for longitudinal or cross-sectional comparisons in different populations [78]. It is evident that more than 40% of overweight children will become obese as adults [79]. Public policies and interventions should be implemented to support school-aged children to participate in exercise programs to improve their physical fitness and decrease obesity levels. In the present study, BMI was significantly correlated with handgrip strength in children of both sexes, and these findings agree with other studies [80]. Obese and overweight children (boys and girls) showed higher upper body muscular strength (handgrip strength) compared to normal weight or underweight (Table 2). This may be attributed to the fact that children with a higher body mass may have greater muscle mass and, therefore, produce a significantly higher amount of force than children with a lower body mass [31]. We did not find any differences in handgrip strength across BMI groups compared to other studies [66,81]. Chronological age does not reflect the growth and maturation events; however, males exhibited an increased growth of muscle strength and muscle mass around the age of 13 years old [82]. Unfortunately, we could not evaluate the pubertal status of the participants.

Significant differences were also found for standing long jump, trunk flexor, and extensor endurance between underweight and overweight groups of both sexes. Meanwhile, underweight children had similar performance compared to normal-weight children. The results were consistent with other reported studies following Greek school children populations [31,34,56,82]. The association between BMI and strength–power tests in the early years seems to be a helpful health marker against obesity for later life.

Cardiorespiratory fitness, as determined by the 20 m shuttle running test, was significantly lower in overweight and obese children than normal weight individuals of both sexes, which other studies have shown as well [83,84]. The results of the present study confirm that overweight and obese children are less physically fit [17]. While several other studies indicated childhood overweight and obesity with low physical fitness are associated with cardiovascular risk factors at a later stage [85].

Muscle strength is similar between boys and girls in pre-puberty (7–11 years old) [86]. Girls outperformed boys in the trunk extensor muscles. Normal-weight girls had lower handgrip and aerobic capacity but greater trunk extensor endurance than boys. Similarly, overweight girls had higher trunk endurance and lower aerobic capacity than boys. The only difference between sexes in the obese group was the greater trunk extensor endurance in girls than boys. This discrepancy may be attributed to the similar body mass between boys and girls in the pre-pubertal stage [86,87], added to the fact that boys present a reduced muscle mass in the pre-pubertal stage [88,89] because the effects of circulating androgens and growth hormone are noticeably later at puberty [86]. This observation is also supported by the girls’ height cross-over, which was obvious in the present study and related to pre-puberty [86]. This is why physical education programs should include strength and endurance exercises at school.

This study has several limitations. First, although we had planned to visit 11 schools, due to the COVID-19 school closure, we eventually entered only four. Consequently, the representativeness of the sample for West Attica is uncertain. Second, despite the known correlation between physical fitness and physical activity, we did not assess physical activity in any way. An added limitation is the lack of an a priori power analysis for the appropriate sample size. The sample size post-hoc analysis revealed a power of less than 80% for all tests except the handgrip strength test. Lastly, we did not evaluate other factors which may be confounders, such SES, pubertal status and maturity offset, and sports participation.

## 5. Conclusions

Many studies indicate that children in rural areas are at a higher risk of obesity. This study confirms the role of BMI on muscle and cardiorespiratory fitness, expanding our knowledge of this specific part of society. BMI is a significant determinant of physical performance. A prevalence as high as 40% for being overweight or obese was observed. There is an urgent need for action from policymakers to promote active lifestyle habits as early as elementary school-aged children. There is also a need to change lifestyle habits and obtain practical education classes for families to prevent childhood obesity. Health specialists should encourage obese children to participate in a long period of aerobic and strength-training exercises.

## Figures and Tables

**Table 1 ijerph-19-11476-t001:** Anthropometric characteristics of the participants (mean ± SD, 95%CI).

Variables	Sex	Age Groups (n)				
		8 (30 Boys, 42 Girls)	9 (61 Boys, 61 Girls)	10 (49 Boys, 54 Girls)	11 (38 Boys, 45 Girls)	12 (9 Boys, 10 Girls)
Height (cm)	Boys (95%CI)	134.3 ± 0.05 (131−136)	137.2 ± 0.06 (135−138) ^†^	141.7 ± 0.05 (140−143) ^¶^	148.8 ± 0.06 (146−151) ^§^	154.3 ± 0.06 (149−158)
	Girls (95%CI)	131.9 ± 0.04 (130−133)	136.4 ± 0.06 (134−138)	143.4 ± 0.07 (141−145) ^¶^	150.1 ± 0.07 (147−152) ^§^	155.5 ± 0.05 (151−159)
Sitting height (cm)	Boys (95%CI)	0.69 ± 0.02 (0.68 ± 0.72)	0.70 ± 0.03 (0.70−0.72)	0.72 ± 0.03 (0.71−0.74) ^¶^	0.74 ± 0.03 (0.73−076)	0.77 ± 0.02 (0.71−0.77)
	Girls (95%CI)	0.68 ± 0.02 (0.69−0.72)	0.69 ± 0.03 (0.68−0.70) ^§,‡^	0.73 ± 0.03 (0.70−0.72)	0.76 ± 0.03 (0.72−0.75)	0.78 ± 0.03 (0.70−0.75)
Weight (kg)	Boys (95%CI)	35.28 ± 8.16 (31.98−38.58)	36.30 ± 8.42 (34.16−38.48)	41.60 ± 10.78 (38.56−44.75)	46.10 ± 10.79 (43.01−49.56)	51.70 ± 12.03 (43.53−57.42)
	Girls (95%CI)	31.40 ± 7.78 (29.00−33.91)	35.10 ± 8.98 (32.76−37.44) ^†^	42.10 ± 10.51 (39.23−45.09) ^¶^	46.3 ± 10.78 (43.01−49.56) ^¶§^	50.4 ± 9.71 (43.53−57.42)
BMI (kg/m^2^)	Boys (95%CI)	19.36 ± 3.13 (18.09−20.62)	19.0 ± 3.16 (18.26−19.88)	20.50 ± 4.32 (19.29−21.77)	20.60 ± 3.75 (19.42−21.89)	21.6 ± 4.38 (18.23−24.97)
	Girls (95%CI)	17.9 ± 3.50 (16.82−19.03)	18.60 ± 3.92 (17.64−19.69)	20.30 ± 4.12 (19.17−21.47)	20.50 ± 4.89 (19.10−22.07)	20.70 ± 2.86 (18.65−22.74)

Values are mean ± standard deviation from 399 participants. ^†^ Significant difference between age group 8 and 9 years old (*p* < 0.05). ^¶^ Significant difference between age group 9 and 10 years old (*p* < 0.01). ^§^ Significant difference between age group 10 and 11 years old (*p* < 0.01). ^‡^ Significant difference between boys and girls (*p* < 0.01).

**Table 2 ijerph-19-11476-t002:** Physical performance test across body mass index groups.

Variables	Sex	UNDERWEIGHT (1 Boy, 6 Girls)	Normal (97 Boys, 110 Girls)	Overweight (40 Boys, 41 Girls)	Obese (49 Obese, 49 Girls)	S1 (98 Boys, 116 Girls)	S2 (89 Boys, 90 Girls)	η^2^ between BMI Groups	r^β^ between S1 and S2
Handgrip Dominant Hand, kg (95%CI)	Boys	11	16.91 ± 4.24 (15.86−17.95)	16.83 ± 3.88 (15.18−18.47)	19.06 ± 4.61 (17.37−20.76) ^†¶^	16.90 ± 4.32 (15.88−17.92)	18.08 ± 4.4 (16.84-19.31)	0.063	0.032
	Girls	12.5 ± 2.91 (10.45−14.55)	15.08 ± 4.42 (13.83−16.32) ^≠^	16.6 ± 5.79 (14.18−19.07)	19.3 ± 4.65 (17.63−20.99) ^¶^	14.91 ± 4.4 (13.76−16.07)	18.4 ± 5.22 (16.92-19.80) ^*^	0.095	0.207
Handgrip Non-dominant Hand, kg (95% CI)	Boys	13	15.98 ± 3.65 (15.08−16.88)	15.87 ± 3.89 (14.23−17.51)	18.29 ± 4.50 (16.63−19.94) ^¶^	16.07 ± 4.01 (15.12−17.02)	17.15 ± 4.00 (16.03-18.30)	0.061	0.052
	Girls	12.5 ± 1.91 (9.45−15.54)	14.68 ± 4.40 (13.44−15.92) ^≠^	15.95 ± 5.12 (13.79−18.12)	18.09 ± 4.65 (16.41−19.77) ^¶^	14.65 ± 4.20 (13.55−15.75)	17.13 ± 5.2 (15.70-18.56) ^*^	0.069	0.177
Long Jump, cm (95%CI)	Boys	118	136.7 ± 24.05 (130.53−142.28)	126.7 ± 13.03 (126.7−137.71)	119.7± 18.51 (112.95−126.53) ^¶^	132.16 ± 22.2 (126.9−137.4)	129.8 ± 21.3 (123.84-135.84) ^*^	0.121	0.157
	Girls	119.5 ± 28.48 (74.16−164.83)	128.3 ± 20.67 (122.45−134.09)	118.5 ± 18.33 (110.8−126.28)	114.5 ± 24.17 (105.8−123.2) ^¶^	118.3 ± 20.29 (112.9−123.6)	124.90 ± 23.93 (118.34-131.53)	0.083	0.015
Trunk Flexors, s (95% CI)	Boys	240	187.9 ± 63.54 (172.3−203.5)	148.04 ± 66.86 (119.8−176.3) ^§^	100.8 ± 63.53 (77.57−124.04) ^¶^	183.7 ± 63.70 (168.6−198.8)	123.04 ± 72.3 (102.7-143.4) ^*^	0.229	0.443
	Girls	240 (240−240)	173.7 ± 71.29 (153.69−193.80)	167.8 ± 69.45 (138.5−197.2)	130.9 ± 74.9 (103.90−157.91) ^¶,≠^	181.6 ± 70.80 (162.9−200.2)	141.7 ± 72.3 (121.8-161.63) ^*^	0.143	0.300
Trunk Extensors, s (95% CI)	Boys	201	142.5 ± 60.66 (127.63−157.46)	94.42 ± 43.71 (75.9−112.8) ^§^	80.45 ± 45.5 (63.7−97.2)^¶^	124.07 ± 61.2 (109.6−138.56)	109.2 ± 60.04 (92.13-125.91)	0.158	0.133
	Girls	171.2 ± 72.02 (56.65−285.85)	162.8 ± 67.96 (143.77−182.00) ^≠^	126.17 ± 61.88 (100.03−152.3) ^≠^	119.84 ± 50.15 (101.7−137.9) ^†,¶,‡,≠^	143.30 ± 71.90 (124.43−162.26)^†^	140.6 ± 56.46 (125.11-156.13)	0.111	0.071
Shuttle run, stages (95% CI)	Boys	31.0	39.8 ± 17.49 (35.58−44.18)	27.6 ± 13.14 (21.96−33.29) ^§^	20.58 ± 13.70 (15.56−25.60) ^¶^	33.15 ± 16.9 (29.1−37.15)	31.5 ± 19.06 (26.21-36.93)	0.214	0.080
	Girls	31.7 ± 10.72 (14.69−48.81)	28.65 ± 12.25 (25.2−32.09) ^≠^	20.4 ± 8.82 (16.32−23.77) ^§^	19.2 ± 8.93 (6.03−22.47) ^¶^	23.14 ± 9.6 (20.6−25.7)	24.5 ± 13.3 (20.83-28.15)	0.202	0.007

Values are mean ± standard deviation from 399 participants (95% CI). S1: subgroup 1; a combined group of underweight and normal weight and S2: subgroup 2, a combined group of overweight and obese. **^†^** Significant difference between overweight and obese (*p* < 0.05). **^¶^** Significant difference between normal weight and obese (*p* < 0.05). **^§^** Significant difference between overweight and normal weight (*p* < 0.05). **^‡^** Significant difference between underweight and obese (*p* < 0.05). **^≠^** Significant difference between boys and girls (*p* < 0.05). ***** Significant difference between S1 and S2 subgroups (*p* < 0.05).

**Table 3 ijerph-19-11476-t003:** Spearman correlations (*p*-values) between BMI and physical performance test.

BMI	Handgrip Dominant Hand	Handgrip Non-Dominant Hand	Long Jump	Trunk Flexors	Trunk Extensors	Shuttle Run
Boys *	0.372	0.370	−0.249	−0.456	−0.301	−0.388
Girls *	0.445	0.394	−0.229	−0.532	−0.249	−0.376
Total *	0.416	0.442	−0.206	−0.403	−0.280	−0.394

* All *p*-values were ≤0.001.

**Table 4 ijerph-19-11476-t004:** Physical performance percentiles for boys (n = 187).

Age (Years)	Test	Percentile (Boys)				
		10	25	50	75	90
8	Handgrip—Dominant (kg)	11.8	13	14	16	17
	Handgrip—Non-dominant (kg)	8.8	11	13	14	17.2
	Long jump (cm)	99.7	103.25	118.5	135	145.2
	Trunk flexors (s)	66.3	81	145	171.25	240
	Trunk extensors (s)	42.8	65.25	87	121.5	152.8
	Shuttle run (stages)	12	13.25	17.5	30.75	39.7
9	Handgrip—Dominant (kg)	11.8	13	15	18	21
	Handgrip—Non-dominant (kg)	12	13	15	17	19
	Long jump (cm)	97.4	109.5	125	131.5	142
	Trunk flexors (s)	61.6	105.5	133	240	240
	Trunk extensors (s)	64.8	74.75	101	141.75	186.9
	Shuttle run (stages)	10.4	21.5	29	41	50
10	Handgrip—Dominant (kg)	12.7	16	18	21	23
	Handgrip—Non-dominant (kg)	13	14	17.5	20	22.3
	Long jump (cm)	105.1	120.75	129.5	140.25	163.9
	Trunk flexors (s)	49.2	77.5	132	240	240
	Trunk extensors (s)	69.8	95	129	185	240
	Shuttle run (stages)	10.7	17.5	33	46.5	53.5
11	Handgrip—Dominant (kg)	15.4	17	19.5	22.75	27
	Handgrip—Non-dominant (kg)	13.7	16	18	21.75	24.3
	Long jump (cm)	119.1	128.25	139.5	149.5	171.3
	Trunk flexors (s)	64.2	102	147.5	240	240
	Trunk extensors (s)	49	81.25	117	139.5	240
	Shuttle run (stages)	13.9	21.5	30	48.75	57.7
12	Handgrip—Dominant (kg)	20	20	21	23	32.2
	Handgrip—Non-dominant (kg)	18.8	20	21	25	29
	Long jump (cm)	140	143	158	160	174.6
	Trunk flexors (s)	57.6	66	240	240	240
	Trunk extensors (s)	37.2	66	67	240	240
	Shuttle run (stages)	13.7	21.5	34.5	56.5	77.8

**Table 5 ijerph-19-11476-t005:** Physical performance percentiles for girls (n = 212).

Age (Years)	Test	Percentile (Boys)				
		10	25	50	75	90
8	Handgrip—Dominant (kg)	10	11	12	14	16
	Handgrip—Non-dominant (kg)	10	11	12	15	16
	Long jump (cm)	91.1	98.3	110	127	135
	Trunk flexors (s)	92.6	125	167.5	240	240
	Trunk extensors (s)	57.5	92.5	137	186.3	240
	Shuttle run (stages)	10.6	17	20	24	35.8
9	Handgrip—Dominant (kg)	11	13	15	17	19
	Handgrip—Non-dominant (kg)	10	11	14	17	18
	Long jump (cm)	94.6	108.5	125	138.5	146.2
	Trunk flexors (s)	42.8	76.5	135	240	240
	Trunk extensors (s)	64.8	83	123.5	198	240
	Shuttle run (stages)	10	14	23.5	28	34.1
10	Handgrip—Dominant (kg)	13	15	17	20	22.7
	Handgrip—Non-dominant (kg)	13	14.25	16	18	20
	Long jump (cm)	104.1	114.75	125.5	140.25	159.7
	Trunk flexors (s)	82.1	112.5	160	240	240
	Trunk extensors (s)	65	95.5	134	210	240
	Shuttle run (stages)	12	15	22	35	46
11	Handgrip—Dominant (kg)	14	17	20	23	27
	Handgrip—Non-dominant (kg)	14.4	17	19	22	24.6
	Long jump (cm)	101.6	116.5	137	152.75	163
	Trunk flexors (s)	73	122	162	240	240
	Trunk extensors (s)	111.3	125.5	187.5	240	240
	Shuttle run (stages)	13	21	28	34	42
12	Handgrip—Dominant (kg)	18.8	19	21	27	30.8
	Handgrip—Non-dominant (kg)	18.8	21	23	27	31
	Long jump (cm)	88	140	159	175.5	179.2
	Trunk flexors (s)	156.4	240	240	240	240
	Trunk extensors (s)	125.8	167.5	225	240	240
	Shuttle run (stages)	21	33	50	66	66

## Data Availability

The data that support the findings of this study are available from the corresponding author, C.T., upon reasonable request.

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
