# Peer review of "The Impact of Obesity on the Fitness Performance of School-Aged Children Living in Rural Areas—The West Attica Project"

_ijerph, 2022, doi:10.3390/ijerph191811476_

Round 1
Reviewer 1 Report
Your study is very interesting presenting valuable information about a significant issue of children's health. When its shortcomings are addressed, a nice manuscript will be given to the readers of the IJERPH. Please, check the comments below, as well as those in the attached pdf.
Introduction
There is a strong (but unnecessary) focus on children’s physical activity (PA) and the association between PA and BMI. Although fitness is strongly associated with PA, these two factors are not interchangeable. Moreover, in this study participants’ PA was not examined.
The introduction should be reformed to lead the reader to the aim of the study through the presentation of relevant research findings. A proposed “roadmap” for this could be the following:
· Prevalence of obesity (worldwide/in Greece)
· The impact of obesity on children’s health (general topic)
· The importance of physical fitness for children’s health
· The impact of obesity on children’s physical fitness
· Physical fitness, BMI and their association in children who live in rural areas, trying to give a rationale why those children have (or do not have) lower levels on physical fitness and higher BMI.
A short picture of west Attica should be provided. This would help the reader to understand why children from West Attica are examined and provide some rationale for presenting how family’s SES can have an impact on children’s physical fitness.
Methods
1. The authors should provide information regarding:
· - how were the participants recruited
· -what was the rate of participation agreement
· - how were the participants examined (individually or in groups? how many children per group?)
· -inter-rater reliability
· - when the study took place
· - the technical adequacy of the tests used.
· -the subgroups (S1, S2) created
2. Please, calculate the effect sizes for both the differences and the correlations examined. Statistical significance does not give the whole picture.
Results
Please, provide the size of subgroups of this study
Discussion
Similarly with the Introduction, emphasis is given on the association between BMI and PA. However, in this study children’s PA is not examined; thus, this strong emphasis is not needed nor justified.
The limitations of the study should be presented.
Conclusions
The conclusions should be re-written to reflect the main findings of the study.

Author Response
Dear reviewer,
Thank you very much for your precious and detailed review of our article.
Taking into consideration your suggestions, we have the following:
Introduction
We followed your suggestions and revised almost the whole introduction based on your recommended roadmap. So, we would appreciate your time to make us any new comments on this new introduction section.
Methods
1. See sections "2.1. Participants" and "2.2. Procedure" for:
- how were the participants recruited
- how were the participants examined (individually or in groups? how many children per group?)
- when the study took place
- the technical adequacy of the tests use
And section "2.3. Data Analysis" for the subgroups (S1, S2) as determined.
About "what was the rate of the participation agreement," we don't have data about it.
The inter-rater reliability did not examine.
We also took into consideration the rest of your suggestions inside the manuscript.
Results
We provided the size of each subgroup in the tables.
Discussion
The discussion section was revised based on the introduction and your suggestions.
As you requested, the limitations of the study are presented in the last paragraph of the discussion (lines 453-461).
You will also find more corrections on the manuscript based on the other reviewers' comments. More statistical analyses were conducted, such as regression analysis and calculation of the power of the sample size. Regarding regression analysis, we chose to run hierarchical linear regression, putting on Model 1 the BMI (our primary independent variable) and Model 2 and 3 the age and sex, respectively. For post-hoc power analysis of the sample size, we used G*Power 3.1.9.7.
Thanks for your insightful suggestions and comments. We hope that all are covered in our new manuscript.
Best regards
Reviewer 2 Report
Thank you for inviting me to review this paper, which investigates the relationship between body mass index (BMI) and muscle /cardiorespiratory fitness in in Greece children ages 8-12 years old. The above-mentioned relationship has been examined extensively, and extended the results to younger populations in Greece. This study could make a useful contribution in an area where the evidence is still unclear (lifestyle among children and prevalence of overweight/obesity ). While I believe this is an important area of study, I have offered suggestions/comments on how to improve the manuscript.
Introduction
1) Although the study highlights the research gap among children in Greece, the objective could be extended to include covariates such as age, sex…, as the hypothesis is very obvious and does not involve any novelty. Since BMI is related to the fitness status of children (but not adults who are very muscular) and has been sufficiently studied, the aim of the study was to provide further new insights. In this form, the hypothesis seems very weak. Since the introduction and the discussion talk about the development of overweight in different regions in Greece, perhaps a national cross-sectional comparison would be of interest. Perhaps you have also collected other covariates that could be relevant in this context, e.g. social status?
2) I am not sure whether "gender" or "sex" is the better choice. Many journals require a distinction.
Methods
1) The method section is clearly described and easy to understand. However, it was not clear to me where exactly the data collection took place and whether it involved different PE teachers or study staff (allowing for the possibility of Risk of Bias).
2)Recently it has become common to do regression analyses instead of correlations. Perhaps this makes more use in your evaluation. Linear regression models could determine the relevant predictors (age, sex, socioeconomic status, physical fitness). It is unfortunate to hide them for analysis.
3) Since you are using the Kruskal-Wallis test (and other tests for group comparisons: BMI, gender, age), it would be advisable to improve the quality of the study and calculate the power of the sample size (selection bias is defined unclear when power of sample size is not calculated).
Results
Please follow the comments in the methods.
Table 1 and Table 2: Please add the number of participants for individual group comparisons (between age group and BMI)
Discussion
The discussion reads well. If they take into account the comments above, there may be room for discussion.
Line 240: You can not confirm that the children had a low physical activity level! You did not mesuared it. This is pure speculation. Please tone down the sentence.
Minor comments:
Line 24: There“ was“ negative… since you use the past tense in the text
Line 57… Gender ??? or sex???
Line 69-70 sentence repeats itself
Line 76-78 I miss the reference for this data: Similarly, children and adoles-76 cents ages 6-18 years old living on Greek islands also have a higher prevalence…
Author Response
Dear reviewer,
Thank you very much for your precious and detailed review of our article.
Taking into consideration your suggestions, we have the following:
Introduction
1) We did not collect any other data, like social status. See "regression analysis" (lines 232-236) for all variables we collected.
2) As we divided children by their biological characteristics, we changed "gender" to "sex" on the whole manuscript.
Please notice that we revised almost the whole introduction section, as one reviewer recommended.
Methods
1) Added in lines 137-138
2) As you recommended, we did the regression analysis. We chose to run hierarchical linear regression, putting on Model 1 the BMI (our primary independent variable) and Model 2 and 3 the age and sex, respectively.
3) Power of the sample size was calculated. We used G*Power 3.1.9.7 for Windows.
Results
We followed the comments from the method and added the number of participants for individual group comparisons. We also present the results of the regression analysis.
Discussion
We changed a lot of sentences, and we added much new information.
Line 240: We changed it to "low physical performance," as our initial hypothesis was.
Minor comments: all suggestions were corrected.
Thanks for your insightful suggestions and comments. We hope that all are covered in our new manuscript.
Best regards
Reviewer 3 Report
Dear authors, the topic of the work is current and represents one of the burning issues for the complete population, especially the youngest population. The work itself is presented adequately and interestingly and can serve as a basis for further research in the future. Also, I noticed a few minor bugs that you should correct. First, in line 20 the number of girls is missing and in line 109 there is a lack of an enumeration of weights.
I wish you much success in your future work.
Author Response
Dear reviewer,
Thanks for your feedback and your valuable comments.
As you recommended, we added the sample number of the girls in line 20 (21 on the new manuscript). However, we are unsure what you mean in line 109 with "enumeration of weights." We have already written the nearest number of each unit (e.g., 0.1 kg for the weight) as we measured. Please inform us if you suggest something more.
Please notice that we revised almost the whole introduction section, as one reviewer recommended. Also, we have conducted added statistical analysis tests (post-hoc power analysis, regression analysis, and effect size estimation). The discussion section has been revised accordingly.
Best regards
Round 2
Reviewer 1 Report
The authors have revised the manuscript according to reviewers' comments and have successfully addressed all the shortcomings that had been noticed.
Author Response
Dear reviewer,
Thanks for your positive feedback and the careful reading of our manuscript.
Best regards,
The Authors